# Predicting Organic Reaction Outcomes with Weisfeiler-Lehman Network

**Wengong Jin**[†]    **Connor W. Coley**[‡]    **Regina Barzilay**[†]    **Tommi Jaakkola**[†]
[†]Computer Science and Artificial Intelligence Lab, MIT
[‡]Department of Chemical Engineering, MIT
[†]{wengong,regina,tommi}@csail.mit.edu, [‡]ccoley@mit.edu

## Abstract

The prediction of organic reaction outcomes is a fundamental problem in computational chemistry. Since a reaction may involve hundreds of atoms, fully exploring the space of possible transformations is intractable. The current solution utilizes reaction templates to limit the space, but it suffers from coverage and efficiency issues. In this paper, we propose a template-free approach to efficiently explore the space of product molecules by first pinpointing the reaction center – the set of nodes and edges where graph edits occur. Since only a small number of atoms contribute to reaction center, we can directly enumerate candidate products. The generated candidates are scored by a Weisfeiler-Lehman Difference Network that models high-order interactions between changes occurring at nodes across the molecule. Our framework outperforms the top-performing template-based approach with a 10% margin, while running orders of magnitude faster. Finally, we demonstrate that the model accuracy rivals the performance of domain experts.

## 1   Introduction

One of the fundamental problems in organic chemistry is the prediction of which products form as a result of a chemical reaction [16, 17]. While the products can be determined unambiguously for simple reactions, it is a major challenge for many complex organic reactions. Indeed, experimentation remains the primary manner in which reaction outcomes are analyzed. This is time consuming, expensive, and requires the help of an experienced chemist. The empirical approach is particularly limiting for the goal of automatically designing efficient reaction sequences that produce specific target molecule(s), a problem known as chemical retrosynthesis [16, 17].

Viewing molecules as labeled graphs over atoms, we propose to formulate the reaction prediction task as a graph transformation problem. A chemical reaction transforms input molecules (reactants) into new molecules (products) by performing a set of graph edits over reactant molecules, adding new edges and/or eliminating existing ones. Given that a typical reaction may involve more than 100 atoms, fully exploring all possible transformations is intractable. The computational challenge is how to reduce the space of possible edits effectively, and how to select the product from among the resulting candidates.

The state-of-the-art solution is based on *reaction templates* (Figure 1). A reaction template specifies a molecular subgraph pattern to which it can be applied and the corresponding graph transformation. Since multiple templates can match a set of reactants, another model is trained to filter candidate products using standard supervised approaches. The key drawbacks of this approach are coverage and scalability. A large number of templates is required to ensure that at least one can reconstitute the correct product. The templates are currently either hand-crafted by experts [7, 1, 15] or generated from reaction databases with heuristic algorithms [2, 11, 3]. For example, Coley et al. [3] extracts 140K unique reaction templates from a database of 1 million reactions. Beyond coverage, applying a

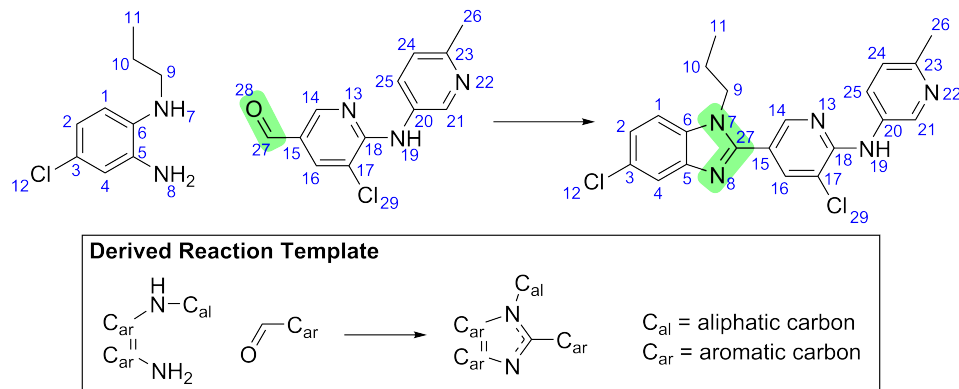

Figure 1: An example reaction where the reaction center is (27,28), (7,27), and (8,27), highlighted in green. Here bond (27,28) is deleted and (7,27) and (8,27) are connected by aromatic bonds to form a new ring. The corresponding reaction template consists of not only the reaction center, but nearby functional groups that explicitly specify the context.

template involves graph matching and this makes examining large numbers of templates prohibitively expensive. The current approach is therefore limited to small datasets with limited types of reactions.

In this paper, we propose a template-free approach by learning to identify the *reaction center*, a small set of atoms/bonds that change from reactants to products. In our datasets, on average only 5.5% of the reactant molecules directly participate in the reaction. The small size of the reaction centers together with additional constraints on bond formations enables us to directly enumerate candidate products. Our forward-prediction approach is then divided into two key parts: (1) learning to identify reaction centers and (2) learning to rank the resulting enumerated candidate products.

Our technical approach builds on neural embedding of the *Weisfeiler-Lehman* isomorphism test. We incorporate a specific attention mechanism to identify reaction centers while leveraging distal chemical effects not accounted for in related convolutional representations [5, 4]. Moreover, we propose a novel *Weisfeiler-Lehman Difference Network* to learn to represent and efficiently rank candidate transformations between reactants and products.

We evaluate our method on two datasets derived from the USPTO [13], and compare our methods to the current top performing system [3]. Our method achieves 83.9% and 77.9% accuracy on two datasets, outperforming the baseline approach by 10%, while running 140 times faster. Finally, we demonstrate that the model outperforms domain experts by a large margin.

## 2   Related Work

**Template-based Approach**  Existing machine learning models for product prediction are mostly built on reaction templates. These approaches differ in the way templates are specified and in the way the final product is selected from multiple candidates. For instance, Wei et al. [18] learns to select among 16 pre-specified, hand-encoded templates, given fingerprints of reactants and reagents. While this work was developed on a narrow range of chemical reaction types, it is among the first implementations that demonstrates the potential of neural models for analyzing chemical reactions.

More recent work has demonstrated the power of neural methods on a broader set of reactions. For instance, Segler and Waller [14] and Coley et al. [3] use a data-driven approach to obtain a large set of templates, and then employ a neural model to rank the candidates. The key difference between these approaches is the representation of the reaction. In Segler and Waller [14], molecules are represented based on their Morgan fingerprints, while Coley et al. [3] represents reactions by the features of atoms and bonds in the reaction center. However, the template-based architecture limits both of these methods in scaling up to larger datasets with more diversity.

**Template-free Approach**  Kayala et al. [8] also presented a template-free approach to predict reaction outcomes. Our approach differs from theirs in several ways. First, Kayala et al. operates at the mechanistic level - identifying elementary mechanistic steps rather than the overall transformations from reactants to products. Since most reactions consist of many mechanistic steps, their approach

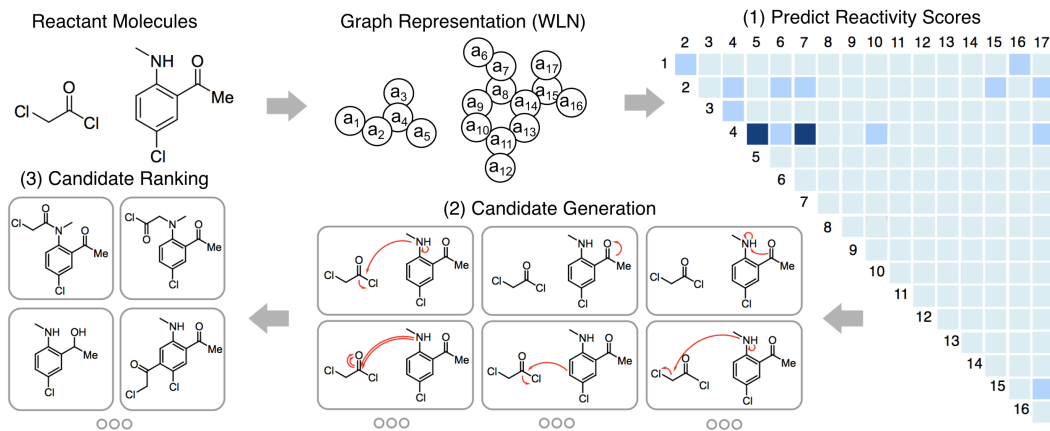

Figure 2: Overview of our approach. (1) we train a model to identify pairwise atom interactions in the reaction center. (2) we pick the top $K$ atom pairs and enumerate chemically-feasible bond configurations between these atoms. Each bond configuration generates a candidate outcome of the reaction. (3) Another model is trained to score these candidates to find the true product.

requires multiple predictions to fulfill an entire reaction. Our approach operates at the graph level - predicting transformations from reactants to products in a single step. Second, mechanistic descriptions of reactions are not given in existing reaction databases. Therefore, Kayala et al. created their training set based on a mechanistic-level template-driven expert system. In contrast, our model is learned directly from real-world experimental data. Third, Kayala et al. uses feed-forward neural networks where atoms and graphs are represented by molecular fingerprints and additional hand-crafted features. Our approach builds from graph neural networks to encode graph structures.

**Molecular Graph Neural Networks** The question of molecular graph representation is a key issue in reaction modeling. In computational chemistry, molecules are often represented with Morgan Fingerprints, boolean vectors that reflect the presence of various substructures in a given molecule. Duvenaud et al. [5] developed a neural version of Morgan Fingerprints, where each convolution operation aggregates features of neighboring nodes as a replacement of the fixed hashing function. This representation was further expanded by Kearnes et al. [9] into graph convolution models. Dai et al. [4] consider a different architecture where a molecular graph is viewed as a latent variable graphical model. Their recurrent model is derived from Belief Propagation-like algorithms. Gilmer et al. [6] generalized all previous architectures into message-passing network, and applied them to quantum chemistry. The closest to our work is the Weisfeiler-Lehman Kernel Network proposed by Lei et al. [12]. This recurrent model is derived from the Weisfeiler-Lehman kernel that produces isomorphism-invariant representations of molecular graphs. In this paper, we further enhance this representation to capture graph transformations for reaction prediction.

## 3 Overview

Our approach bypasses reaction templates by learning a *reaction center identifier*. Specifically, we train a neural network that operates on the reactant graph to predict a reactivity score for every pair of atoms (Section 3.1). A reaction center is then selected by picking a small number of atom pairs with the highest reactivity scores. After identifying the reaction center, we generate possible product candidates by enumerating possible bond configurations between atoms in the reaction center (Section 3.2) subject to chemical constraints. We train another neural network to rank these product candidates (represented as graphs, together with the reactants) so that the correct reaction outcome is ranked highest (Section 3.3). The overall pipeline is summarized in Figure 2. Before describing the two modules in detail, we formally define some key concepts used throughout the paper.

**Chemical Reaction** A chemical reaction is a pair of molecular graphs $(G_r, G_p)$, where $G_r$ is called the *reactants* and $G_p$ the *products*. A molecular graph is described as $G = (V, E)$, where $V = \{a_1, a_2, \cdots, a_n\}$ is the set of atoms and $E = \{b_1, b_2, \cdots, b_m\}$ is the set of associated bonds of varying types (single, double, aromatic, etc.). Note that $G_r$ is has multiple connected components

since there are multiple molecules comprising the reactants. The reactions used for training are *atom-mapped* so that each atom in the product graph has a unique corresponding atom in the reactants.

**Reaction Center** A reaction center is a set of atom pairs $\{(a_i, a_j)\}$, where the bond type between $a_i$ and $a_j$ differs from $G_r$ to $G_p$. In other words, a reaction center is a minimal set of *graph edits* needed to transform reactants to products. Since the reported reactions in the training set are atom-mapped, reaction centers can be identified automatically given the product.

## 3.1 Reaction Center Identification

In a given reaction $R = (G_r, G_p)$, each atom pair $(a_u, a_v)$ in $G_r$ is associated with a reactivity label $y_{uv} \in \{0, 1\}$ specifying whether their relation differs between reactants and products. The label is determined by comparing $G_r$ and $G_p$ with the help of atom-mapping. We predict the label on the basis of learned atom representations that incorporate contextual cues from the surrounding chemical environment. In particular, we build on a Weisfeiler-Lehman Network (WLN) that has shown superior results against other learned graph representations in the narrower setting of predicting chemical properties of individual molecules [12].

### 3.1.1 Weisfeiler-Lehman Network (WLN)

The WLN is inspired by the Weisfeiler-Lehman isomorphism test for labeled graphs. The architecture is designed to embed the computations inherent in WL isomorphism testing to generate learned isomorphism-invariant representations for atoms.

**WL Isomorphism Test** The key idea of the isomorphism test is to repeatedly augment node labels by the sorted set of node labels of neighbor nodes and to compress these augmented labels into new, short labels. The initial labeling is the atom element. In each iteration, its label is augmented with the element labels of its neighbors. Such a multi-set label is compactly represented as a new label by a hash function. Let $c_v^{(L)}$ be the final label of atom $a_v$. The molecular graph $G = (V, E)$ is represented as a set $\{(c_u^{(L)}, b_{uv}, c_v^{(L)}) \mid (u, v) \in E\}$, where $b_{uv}$ is the bond type between $u$ and $v$. Two graphs are said to be isomorphic if their set representations are the same. The number of distinct labels grows exponentially with the number of iterations $L$.

**WL Network** The discrete relabeling process does not directly generalize to continuous feature vectors. Instead, we appeal to neural networks to continuously embed the computations inherent in the WL test. Let $r$ be the analogous continuous relabeling function. Then a node $v \in G$ with neighbor nodes $N(v)$, node features $\mathbf{f}_v$, and edge features $\mathbf{f}_{uv}$ is "relabeled" according to

$$r(v) = \tau(\mathbf{U}_1 \mathbf{f}_v + \mathbf{U}_2 \sum_{u \in N(v)} \tau(\mathbf{V}[\mathbf{f}_u, \mathbf{f}_{uv}])) \tag{1}$$

where $\tau(\cdot)$ could be any non-linear function. We apply this relabeling operation iteratively to obtain context-dependent atom vectors

$$\mathbf{h}_v^{(l)} = \tau(\mathbf{U}_1 \mathbf{h}_v^{(l-1)} + \mathbf{U}_2 \sum_{u \in N(v)} \tau(\mathbf{V}[\mathbf{h}_u^{(l-1)}, \mathbf{f}_{uv}])) \qquad (1 \leq l \leq L) \tag{2}$$

where $\mathbf{h}_v^{(0)} = \mathbf{f}_v$ and $\mathbf{U}_1, \mathbf{U}_2, \mathbf{V}$ are shared across layers. The final atom representations arise from mimicking the set comparison function in the WL isomorphism test, yielding

$$\mathbf{c}_v = \sum_{u \in N(v)} \mathbf{W}^{(0)} \mathbf{h}_u^{(L)} \odot \mathbf{W}^{(1)} \mathbf{f}_{uv} \odot \mathbf{W}^{(2)} \mathbf{h}_v^{(L)} \tag{3}$$

The set comparison here is realized by matching each rank-1 edge tensor $\mathbf{h}_u^{(L)} \otimes \mathbf{f}_{uv} \otimes \mathbf{h}_v^{(L)}$ to a set of reference edges also cast as rank-1 tensors $\mathbf{W}^{(0)}[k] \otimes \mathbf{W}^{(1)}[k] \otimes \mathbf{W}^{(2)}[k]$, where $\mathbf{W}[k]$ is the $k$-th row of matrix $\mathbf{W}$. In other words, Eq. 3 above could be written as

$$\mathbf{c}_v[k] = \sum_{u \in N(v)} \left\langle \mathbf{W}^{(0)}[k] \otimes \mathbf{W}^{(1)}[k] \otimes \mathbf{W}^{(2)}[k], \quad \mathbf{h}_u^{(L)} \otimes \mathbf{f}_{uv} \otimes \mathbf{h}_v^{(L)} \right\rangle \tag{4}$$

The resulting $\mathbf{c}_v$ is a vector representation that captures the local chemical environment of the atom (through relabeling) and involves a comparison against a learned set of reference environments. The representation of the whole graph $G$ is simply the sum over all the atom representations: $\mathbf{c}_G = \sum_v \mathbf{c}_v$.

### 3.1.2 Finding Reaction Centers with WLN

We present two models to predict reactivity: the *local* and *global* models. Our local model is based directly on the atom representations $\mathbf{c}_u$ and $\mathbf{c}_v$ in predicting label $y_{uv}$. The global model, on the other hand, selectively incorporates distal chemical effects with the goal of capturing the fact that atoms outside of the reaction center may be necessary for the reaction to occur. For example, the reaction center may be influenced by certain *reagents*[1]. We incorporate these distal effects into the global model through an attention mechanism.

**Local Model** Let $\mathbf{c}_u, \mathbf{c}_v$ be the atom representations for atoms $u$ and $v$, respectively, as returned by the WLN. We predict the reactivity score of $(u, v)$ by passing these through another neural network:

$$s_{uv} = \sigma\left(\mathbf{u}^T \tau(\mathbf{M}_a \mathbf{c}_u + \mathbf{M}_a \mathbf{c}_v + \mathbf{M}_b \mathbf{b}_{uv})\right) \tag{5}$$

where $\sigma(\cdot)$ is the sigmoid function, and $\mathbf{b}_{uv}$ is an additional feature vector that encodes auxiliary information about the pair such as whether the two atoms are in different molecules or which type of bond connects them.

**Global Model** Let $\alpha_{uv}$ be the attention score of atom $v$ on atom $u$. The global context representation $\tilde{\mathbf{c}}_u$ of atom $u$ is calculated as the weighted sum of all reactant atoms where the weight comes from the attention module:

$$\tilde{\mathbf{c}}_u = \sum_v \alpha_{uv} \mathbf{c}_v; \qquad \alpha_{uv} = \sigma\left(\mathbf{u}^T \tau(\mathbf{P}_a \mathbf{c}_u + \mathbf{P}_a \mathbf{c}_v + \mathbf{P}_b \mathbf{b}_{uv})\right) \tag{6}$$

$$s_{uv} = \sigma\left(\mathbf{u}^T \tau(\mathbf{M}_a \tilde{\mathbf{c}}_u + \mathbf{M}_a \tilde{\mathbf{c}}_v + \mathbf{M}_b \mathbf{b}_{uv})\right) \tag{7}$$

Note that the attention is obtained with sigmoid rather than softmax non-linearity since there may be multiple atoms relevant to a particular atom $u$.

**Training** Both models are trained to minimize the following loss function:

$$\mathcal{L}(\mathcal{T}) = -\sum_{R \in \mathcal{T}} \sum_{u \neq v \in R} y_{uv} \log(s_{uv}) + (1 - y_{uv}) \log(1 - s_{uv}) \tag{8}$$

Here we predict each label independently because of the large number of variables. For a given reaction with $N$ atoms, we need to predict the reactivity score of $O(N^2)$ pairs. This quadratic complexity prohibits us from adding higher-order dependencies between different pairs. Nonetheless, we found independent prediction yields sufficiently good performance.

## 3.2 Candidate Generation

We select the top $K$ atom pairs with the highest predicted reactivity score and designate them, collectively, as the reaction center. The set of candidate products are then obtained by enumerating all possible bond configuration changes within the set. While the resulting set of candidate products is exponential in $K$, many can be ruled out by invoking additional constraints. For example, every atom has a maximum number of neighbors they can connect to (*valence constraint*). We also leverage the statistical bias that reaction centers are very unlikely to consist of disconnected components (*connectivity constraint*). Some multi-step reactions do exist that violate the connectivity constraint. As we will show, the set of candidates arising from this procedure is more compact than those arising from templates without sacrificing coverage.

## 3.3 Candidate Ranking

The training set for candidate ranking consists of lists $\mathcal{T} = \{(r, p_0, p_1, \cdots, p_m)\}$, where $r$ are the reactants, $p_0$ is the known product, and $p_1, \cdots, p_m$ are other enumerated candidate products. The goal is to learn a scoring function that ranks the highest known product $p_0$. The challenge in ranking candidate products is again representational. We must learn to represent $(r, p)$ in a manner that can focus on the key difference between the reactants $r$ and products $p$ while also incorporating the necessary chemical contexts surrounding the changes.

We again propose two alternative models to score each candidate pair $(r, p)$. The first model naively represents a reaction by summing difference vectors of all atom representations obtained from a WLN on the associated connected components. Our second and improved model, called WLDN, takes into account higher order interactions between these differences vectors.

**WLN with Sum-Pooling** Let $\mathbf{c}_v^{(p_i)}$ be the learned atom representation of atom $v$ in candidate product molecule $p_i$. We define *difference vector* $\mathbf{d}_v^{(p_i)}$ pertaining to atom $v$ as follows:

$$\mathbf{d}_v^{(p_i)} = \mathbf{c}_v^{(p_i)} - \mathbf{c}_v^{(r)}; \qquad s(p_i) = \mathbf{u}^T \tau(\mathbf{M} \sum_{v \in p_i} \mathbf{d}_v^{(p_i)}) \tag{9}$$

Recall that the reactants and products are atom-mapped so we can use $v$ to refer to the same atom. The pooling operation is a simple sum over these difference vectors, resulting in a single vector for each $(r, p_i)$ pair. This vector is then fed into another neural network to score the candidate product $p_i$.

**Weisfeiler-Lehman Difference Network (WLDN)** Instead of simply summing all difference vectors, the WLDN operates on another graph called a *difference graph*. A difference graph $D(r, p_i)$ is defined as a molecular graph which has the same atoms and bonds as $p_i$, with atom $v$'s feature vector replaced by $\mathbf{d}_v^{(p_i)}$. Operating on the difference graph has several benefits. First, in $D(r, p_i)$, atom $v$'s feature vector deviates from zero only if it is close to the reaction center, thus focusing the processing on the reaction center and its immediate context. Second, $D(r, p_i)$ explicates neighbor dependencies between difference vectors. The WLDN maps this graph-based representation into a fixed-length vector, by applying a separately parameterized WLN on top of $D(r, p_i)$:

$$\mathbf{h}_v^{(p_i, l)} = \tau \left( \mathbf{U}_1 \mathbf{h}_v^{(p_i, l-1)} + \mathbf{U}_2 \sum_{u \in N(v)} \tau \left( \mathbf{V}[\mathbf{h}_u^{(p_i, l-1)}, \mathbf{f}_{uv}] \right) \right) \quad (1 \le l \le L) \tag{10}$$

$$\mathbf{d}_v^{(p_i, L)} = \sum_{u \in N(v)} \mathbf{W}^{(0)} \mathbf{h}_u^{(p_i, L)} \odot \mathbf{W}^{(1)} \mathbf{f}_{uv} \odot \mathbf{W}^{(2)} \mathbf{h}_v^{(p_i, L)} \tag{11}$$

where $\mathbf{h}_v^{(p_i, 0)} = \mathbf{d}_v^{(p_i)}$. The final score of $p_i$ is $s(p_i) = \mathbf{u}^T \tau(\mathbf{M} \sum_{v \in p_i} \mathbf{d}_v^{(p_i, L)})$.

**Training** Both models are trained to minimize the softmax log-likelihood objective over the scores $\{s(p_0), s(p_1), \cdots, s(p_m)\}$ where $s(p_0)$ corresponds to the target.

## 4 Experiments

**Data** As a source of data for our experiments, we used reactions from USPTO granted patents, collected by Lowe [13]. After removing duplicates and erroneous reactions, we obtained a set of 480K reactions, to which we refer in the paper as USPTO. This dataset is divided into 400K, 40K, and 40K for training, development, and testing purposes.[2]

In addition, for comparison purposes we report the results on the subset of 15K reaction from this dataset (referred as USPTO-15K) used by Coley et al. [3]. They selected this subset to include reactions covered by the 1.7K most common templates. We follow their split, with 10.5K, 1.5K, and 3K for training, development, and testing.

**Setup for Reaction Center Identification** The output of this component consists of $K$ atom pairs with the highest reactivity scores. We compute the *coverage* as the proportion of reactions where all atom pairs in the true reaction center are predicted by the model, i.e., where the recorded product is found in the model-generated candidate set.

The model features reflect basic chemical properties of atoms and bonds. Atom-level features include its elemental identity, degree of connectivity, number of attached hydrogen atoms, implicit valence, and aromaticity. Bond-level features include bond type (single, double, triple, or aromatic), whether it is conjugated, and whether the bond is part of a ring.

Both our local and global models are build upon a Weisfeiler-Lehman Network, with unrolled depth 3. All models are optimized with Adam [10], with learning rate decay factor 0.9.

| USPTO-15K | | | | |
|---|---|---|---|---|
| **Method** | $|\theta|$ | **K=6** | **K=8** | **K=10** |
| Local | 572K | 80.1 | 85.0 | 87.7 |
| Local | 1003K | 81.6 | 86.1 | 89.1 |
| Global | 756K | **86.7** | **90.1** | **92.2** |
| **USPTO** | | | | |
| Local | 572K | 83.0 | 87.2 | 89.6 |
| Global | 756K | **89.8** | **92.0** | **93.3** |
| **Avg. Num. of Candidates (USPTO)** | | | | |
| Template | - | | 482.3 out of 5006 | |
| Global | - | 60.9 | 246.5 | 1076 |

| USPTO-15K | | | | |
|---|---|---|---|---|
| **Method** | **Cov.** | **P@1** | **P@3** | **P@5** |
| Coley et al. | 100.0 | 72.1 | 86.6 | 90.7 |
| WLN | 90.1 | 74.9 | 84.6 | 86.3 |
| WLDN | 90.1 | **76.7** | **85.6** | **86.8** |
| WLN (*) | 100.0 | 81.4 | 92.5 | 94.8 |
| WLDN (*) | 100.0 | **84.1** | **94.1** | **96.1** |
| **USPTO** | | | | |
| Method | $|\theta|$ | **P@1** | **P@3** | **P@5** |
| WLDN | 3.2M | **79.6** | **87.7** | **89.2** |
| WLDN (*) | 3.2M | 83.9 | 93.2 | 95.2 |

(a) Reaction Center Prediction Performance. Coverage is reported by picking the top $K$ ($K$=6,8,10) reactivity pairs. $|\theta|$ is the number of model parameters.

(b) Candidate Ranking Performance. Precision at ranks 1,3,5 are reported. (*) denotes that the true product was added if not covered by the previous stage.

Table 1: Model Comparison on USPTO-15K and USPTO dataset.

**Setup for Candidate Ranking** The goal of this evaluation is to determine whether the model can select the correct product from a set of candidates derived from reaction center. We first compare model accuracy against the top-performing template-based approach by Coley et al. [3]. This approach employs frequency-based heuristics to construct reaction templates and then uses a neural model to rank the derived candidates. As explained above, due to the scalability issues associated with this baseline, we can only compare on USPTO-15K, which the authors restricted to contain only examples that were instantiated by their most popular templates. For this experiment, we set $K = 8$ for candidate generation, which achieves 90% coverage and yields 250 candidates per reaction. To compare a standard WLN representation against its counterpart with Difference Networks (WLDN), we train them under the same setup on USPTO-15K, fixing the number of parameters to 650K.

Next, we evaluate our model on USPTO for large scale evaluation. We set $K = 6$ for candidate generation and report the result of the best model architecture. Finally, to factorize the coverage of candidate selection and the accuracy of candidate ranking, we consider two evaluation scenarios: (1) the candidate list as derived from reaction center; (2) the above candidate list augmented with the true product if not found. This latter setup is marked with (*).

## 4.1   Results

**Reaction Center Identification** Table 1a reports the coverage of the model as compared to the real reaction core. Clearly, the coverage depends on the number of atom pairs $K$, with the higher coverage for larger values of $K$. These results demonstrate that even for $K = 8$, the model achieves high coverage, above 90%.

The results also clearly demonstrate the advantage of the global model over the local one, which is consistent across all experiments. The superiority of the global model is in line with the well-known fact that reactivity depends on more than the immediate local environment surrounding the reaction center. The presence of certain *functional groups* (structural motifs that appear frequently in organic chemistry) far from the reaction center can promote or inhibit different modes of reactivity. Moreover, reactivity is often influenced by the presence of *reagents*, which are separate molecules that may not directly contribute atoms to the product. Consideration of both of these factors necessitates the use of a model that can account for long-range dependencies between atoms.

Figure 3 depicts one such example, where the observed reactivity can be attributed to the presence of a reagent molecule that is completely disconnected from the reaction center itself. While the local model fails to anticipate this reactivity, the global one accurately predicts the reaction center. The attention map highlights the reagent molecule as the determinant context.

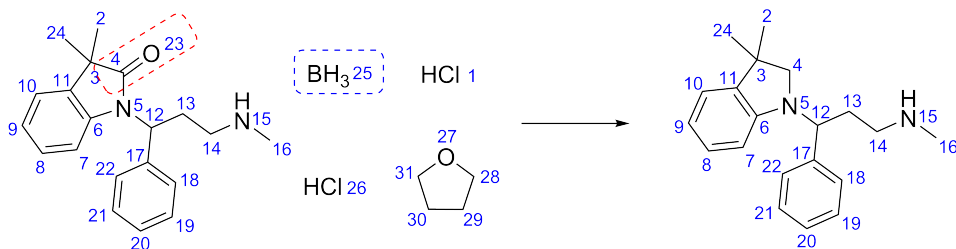

Figure 3: A reaction that reduces the carbonyl carbon of an amide by removing bond 4-23 (red circle). Reactivity at this site would be highly unlikely without the presence of borohydride (atom 25, blue circle). The global model correctly predicts bond 4-23 as the most susceptible to change, while the local model does not even include it in the top ten predictions. The attention map of the global model show that atoms 1, 25, and 26 were determinants of atom 4's predicted reactivity.

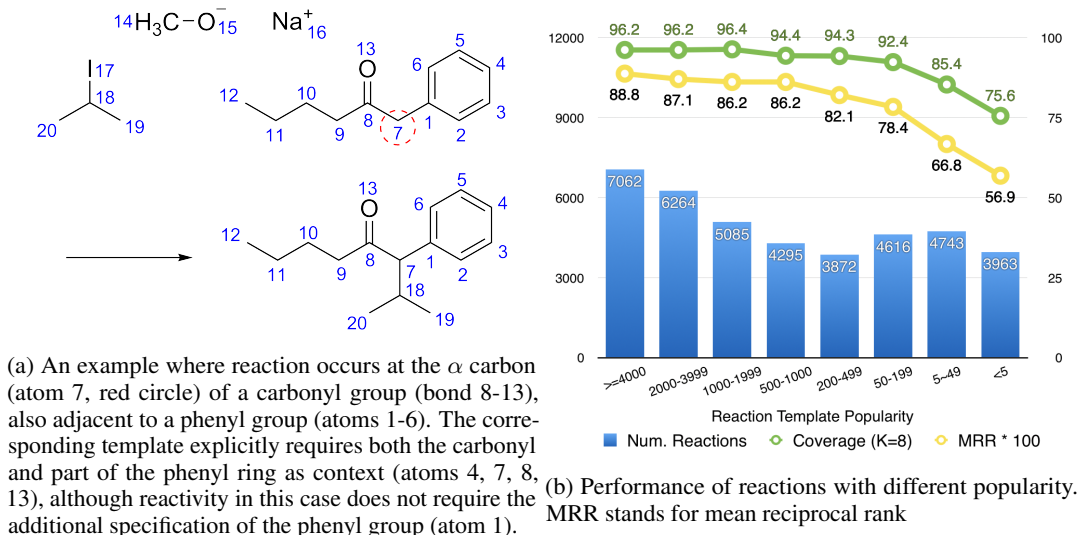

(a) An example where reaction occurs at the $\alpha$ carbon (atom 7, red circle) of a carbonyl group (bond 8-13), also adjacent to a phenyl group (atoms 1-6). The corresponding template explicitly requires both the carbonyl and part of the phenyl ring as context (atoms 4, 7, 8, 13), although reactivity in this case does not require the additional specification of the phenyl group (atom 1).

(b) Performance of reactions with different popularity. MRR stands for mean reciprocal rank

Figure 4

**Candidate Generation** Here we compare the coverage of the generated candidates with the template-based model. Table 1a shows that for $K = 6$, our model generates an average of 60.1 candidates and reaches a coverage of 89.8%. The template-based baseline requires 5006 templates extracted from the training data (corresponding to a minimum of five precedent reactions) to achieve 90.1% coverage with an average of 482 candidates per example.

This weakness of the baseline model can be explained by the difficulty in defining general heuristics with which to extract templates from reaction examples. It is possible to define different levels of specificity based on the extent to which atoms surrounding the reaction center are included or generalized [11]. This introduces an unavoidable trade-off between generality (fewer templates, higher coverage, more candidates) and specificity (more templates, less coverage, fewer candidates). Figure 4a illustrates one reaction example where the corresponding template is rare due to the adjacency of the reaction center to both a carbonyl group and a phenyl ring. Because adjacency to either group can influence reactivity, both are included as part of the template, although reactivity in this case does not require the additional specification of the phenyl group.

The massive number of templates required for high coverage is a serious impediment for the template approach because each template application requires solving a subgraph isomorphism problem. Specifically, it takes on average 7 seconds to apply the 5006 templates to a test instance, while our method takes less than 50 ms, about 140 times faster.

**Candidate Ranking** Table 1b reports the performance on the product prediction task. Since the baseline templates from [3] were optimized on the test and have 100% coverage, we compare its performance against our models to which the correct product is added (WLN(*) and WLDN(*)). Our model clearly outperforms the baseline by a wide margin. Even when compared against the candidates automatically computed from the reaction center, WLDN outperforms the baseline in

top-1 accuracy. The results also demonstrate that the WLDN model consistently outperforms the WLN model. This is consistent with our intuition that modeling higher order dependencies between the difference vectors is advantageous over simply summing over them. Table 1b also shows the model performance improves when tested on the full USPTO dataset.

We further analyze model performance based on the frequency of the underlying transformation as reflected by the the number of template precedents. In Figure 4b we group the test instances according to their frequency and report the coverage of the global model and the mean reciprocal rank (MRR) of the WLDN model on each of them. As expected, our approach achieves the highest performance for frequent reactions. However, it maintains reasonable coverage and ranking accuracy even for rare reactions, which are particularly challenging for template-based methods.

## 4.2 Human Evaluation Study

We randomly selected 80 reaction examples from the test set, ten from each of the template popularity intervals of Figure 4b, and asked ten chemists to predict the outcome of each given its reactants. The average accuracy across the ten performers was 48.2%. Our model achieves an accuracy of 69.1%, very close to the best individual performer who scored 72.0%.

| Chemist | 56.3 | 50.0 | 72.0 | 63.8 | 66.3 | 65.0 | 40.0 | 58.8 | 25.0 | 16.3 |
|---------|------|------|------|------|------|------|------|------|------|------|
| Our Model | | | | | **69.1** | | | | | |

Table 2: Human and model performance on 80 reactions randomly selected from the USPTO test set to cover a diverse range of reaction types. The first 8 are chemists with rich experience in organic chemistry (graduate, postdoctoral and professor level chemists) and the last two are graduate students in chemical engineering who use organic chemistry concepts regularly but have less formal training.

## 5 Conclusion

We proposed a novel template-free approach for chemical reaction prediction. Instead of generating candidate products by reaction templates, we first predict a small set of atoms/bonds in reaction center, and then produce candidate products by enumerating all possible bond configuration changes within the set. Compared to template based approach, our framework runs 140 times faster, allowing us to scale to much larger reaction databases. Both our reaction center identifier and candidate ranking model build from Weisfeiler-Lehman Network and its variants that learn compact representation of graphs and reactions. We hope our work will encourage both computer scientists and chemists to explore fully data driven approaches for this task.

## Acknowledgement

We thank Tim Jamison, Darsh Shah, Karthik Narasimhan and the reviewers for their helpful comments. We also thank members of the MIT Department of Chemistry and Department of Chemical Engineering who participated in the human benchmarking study. This work was supported by the DARPA Make-It program under contract ARO W911NF-16-2-0023.

## Footnotes

[1]Molecules that do not typically contribute atoms to the product but are nevertheless necessary for the reaction to proceed.

[2]Code and data available at https://github.com/wengong-jin/nips17-rexgen

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
