[Supplementary Material · nips-2017-appendix.pdf]

## A  Human Evaluation Setup

Here we describe in detail the human evaluation results in Table 2. The evaluation dataset consists
of eight groups, defined by the reaction template popularity as binned in Figure 4b, each with ten
instances selected randomly from the USPTO test set. We invited in total ten chemists to predict the
product given reactants in all groups.

Table 2: Human and model performance on 80 reactions randomly selected from the USPTO test
set to cover a diverse range of reaction types. The first 8 are experts with rich experience in organic
chemistry (graduate and postdoctoral chemists) and the last two are graduate students in chemical
engineering who use organic chemistry concepts regularly but have less formal training. Our model
performs at the expert chemist level in terms of top 1 accuracy.

| Chemist | 56.3 | 50.0 | 40.0 | 63.8 | 66.3 | 65.0 | 40.0 | 58.8 | 25.0 | 16.3 |
|---|---|---|---|---|---|---|---|---|---|---|
| Our Model | | | | | **69.1** | | | | | |

Figure 5: Details of human performance

(a) Histogram showing the distribution of question
difficulties as evaluated by the average expert perfor-
mance across all ten performers.

(b) Comparison of model performance against human
performance for sets of questions as grouped by the
average human accuracy shown in Figure 5a
.

(c) Individual accuracies of each of the 10 human performers and the WLDN model. Human answers were
scored for exact matches (strict scoring, also reported in Table 2) and imprecise matches (relaxed scoring), where
the difference between the predicted product and the recorded product was small enough to warrant a half-point.
The WLDN model performs at the expert level.