[Reviews · NeurIPS 2017]

Reviewer 1



The paper proposes to model molecular reactions using a Weisfeihler-Lehman graph neural network, an architecture that was previously introduced as a neural network counterpart of the Weisfeihler-Lehman graph kernel. The novelty of the paper resides mainly in the careful application of this neural network framework to chemistry, for predicting reaction centers and ranking chemical reactions. The paper is well written, and most of the neural networks architectural choices for each problem look sound. In WLN+sumpooling, the sum-pooling of differences reduces to the difference of sums, which looses all spatial information. There seems to be an intermediate step of complexity which is therefore missing between the WLN+sumpooling and the WDLN. Applying at least one nonlinear transformation between the difference and pooling operations could have been considered. The authors cite a number of relevant papers both in machine learning and chemistry. To those, one could have also added the original Weisfeiler-Lehman kernel paper as well as some recent papers that use similar graph neural networks (DTNN, MPNN) to predict molecular properties without the bond structure.

Reviewer 2



A deep learning approach is proposed for the application of predicting organic chemical reactions. Given a set of reactants, the algorithm first predicts the likely reaction centers, and then ranks the possible reactions involving those atoms. This paper is well-organized, mostly clear, and makes a significant contribution. Chemical reaction prediction is an important application, and from a machine learning perspective this is a very interesting use of deep learning because of the unique structure of the data --- this paper nicely builds off recent work that uses neural networks to encode graph-structured data. My main comment is that this work is very similar to that of Kayala et. al. (presented at NIPS 2011), who propose a very similar two-step process in which neural networks first predict reaction centers and then rank the predicted products. A number of the details are different in this paper, including the prediction of reaction centers as pairs of atoms rather than separate predictions of electron sources and sinks, the encoded representation of the reaction centers, and the use of a difference graph for the ranking where Kayala et. al. use a Siamese neural network. It would be interesting if the authors could comment on the advantages or disadvantages of these differences. A number of details about the neural network training have been left out, presumably due to space restrictions, but should be included in the appendix at least. These include the network architectures, activation function, initialization, optimization hyperparameters, etc. Learning to Predict Chemical Reactions Matthew A. Kayala, Chloé-Agathe Azencott, Jonathan H. Chen, and Pierre Baldi Journal of Chemical Information and Modeling 2011 51 (9), 2209-2222

Reviewer 3



Summary: This work provides a novel approach to predict the outcome of organic chemical reactions. A reaction can be computationally regarded as graph-prediction problem: given the input of several connected graphs (molecules), the model aims to predict a fully-connected graph (reaction product) that can be obtained by performing several graph edits (reaction) on some edges and nodes (reaction center) in the input graphs. Past reaction predictions involving exhaustively enumeration of reaction centers and fitting them to a large number of existing reaction templates, which is very inefficient and hard to scale. In this work, the author proposed a template-free method to predict the outcome. It is a 3 step pipeline: 1) identify the reaction center given the input graphs using a Weisfeiler-Lehman Network. 2) generate candidate products based their reactivity score and chemical constraints. 3) rank the candidate products using a Weisfeiler-Lehman Difference Network. The proposed method outperformed an existing state-of-art method on a benchmark chemistry dataset in both accuracy (10% rise) and efficiency (140 times faster), and also outperformed human chemist experts. Qualitative Evaluation: Quality: The work is technically sound. The proposed method is well supported by experiments in both real world dataset and human expert comparisons. Clarity: This work describes their work and methods clearly. The experiments are introduced in details. Originality: This work provides a novel solution for reaction outcome prediction, which does not need prior knowledge of reaction templates. The author may want to relate some past NIPS work on computational chemistry to their work: Kayala, Matthew A., and Pierre F. Baldi. "A machine learning approach to predict chemical reactions." Advances in Neural Information Processing Systems. 2011. Significance: The work outperforms the state-of-art of reaction product prediction in both accuracy and efficiency. The user study experiment shows that it also outperforms human experts.